# Phytochemicals in Breast Cancer Prevention and Treatment: A Comprehensive Review

**DOI:** 10.3390/cimb47010030

**Published:** 2025-01-06

**Authors:** Adil Farooq Wali, Jayachithra Ramakrishna Pillai, Sirajunisa Talath, Pooja Shivappa, Sathvik Belagodu Sridhar, Mohamed El-Tanani, Imran Rashid Rangraze, Omnia Ibrahim Mohamed, Nowar Nizar Al Ani

**Affiliations:** 1Department of Pharmaceutical Chemistry, College of Pharmacy, RAK Medical and Health Sciences University, Ras Al Khaimah 11172, United Arab Emirates; sirajunisa@rakmhsu.ac.ae; 2Translational Medicinal Research Centre, Department of Biochemistry, RAK Medical and Health Sciences University, Ras Al Khaimah 11172, United Arab Emirates; poojacs77@gmail.com; 3Department of Clinical Pharmacy & Pharmacology, RAK College of Pharmacy, RAK Medical and Health Sciences University, Ras Al Khaimah 11172, United Arab Emirates; sathvik@rakmhsu.ac.ae; 4RAK College of Pharmacy, RAK Medical and Health Science University, Ras Al Khaimah 11172, United Arab Emirates; eltanani@rakmhsu.ac.ae; 5RAK College of Medical Sciences, RAK Medical and Health Science University, Ras Al Khaimah 11172, United Arab Emirates; imranrashid@rakmhsu.ac.ae; 6Department of General Education, RAK Medical and Health Sciences University, Ras Al Khaimah 11172, United Arab Emirates; omnia@rakmhsu.ac.ae (O.I.M.); nowar@rakmhsu.ac.ae (N.N.A.A.)

**Keywords:** nanotechnology, anticancer activity, flavonoids, apoptosis, signaling pathways inhibition, synergistic effects, targeted drug delivery

## Abstract

Extensive investigation has been conducted on plant-based resources for their pharmacological usefulness, including various cancer types. The scope of this review is wider than several studies with a particular focus on breast cancer, which is an international health concern while studying sources of flavonoids, carotenoids, polyphenols, saponins, phenolic compounds, terpenoids, and glycosides apart from focusing on nursing. Important findings from prior studies are synthesized to explore these compounds’ sources, mechanisms of action, complementary and synergistic effects, and associated side effects. It was reviewed that the exposure to certain doses of catechins, piperlongumine, lycopene, isoflavones and cucurbitacinfor a sufficient period can provide profound anticancer benefits through biological events such as cell cycle arrest, cells undergoing apoptosis and disruption of signaling pathways including, but not limited to JAK-STAT3, HER2-integrin, and MAPK. Besides, the study also covers the potential adverse effects of these phytochemicals. Regarding mechanisms, the widest attention is paid to Complementary and synergistic strategies are discussed which indicate that it would be realistic to alter the dosage and delivery systems of liposomes, nanoparticles, nanoemulsions, and films to enhance efficacy. Future research directions include refining these delivery approaches, further elucidating molecular mechanisms, and conducting clinical trials to validate findings. These efforts could significantly advance the role of phytocompounds in breast cancer management.

## 1. Introduction

### Background

Breast cancer (BC) is recognized to be one of the predominant malignancies found among females, and it is placed as the second most frequent carcinoma over the globe. Because of the influence of various significant factors in emerging the diseased condition, this cancer has been placed under the category of multifactorial disease [1]. According to the statistics and the report from the World Health Organization (WHO), it was perceived that the incidence of breast cancer at a global level is expected to increase in the coming years, specifically by 2030 [2]. Recent studies have indicated that breast cancer survival rates are notably lower in the United Arab Emirates (UAE) and Saudi Arabia compared to countries like Australia and Canada. This disparity has prompted researchers to investigate the key factors associated with breast cancer outcomes in these regions [2].

The differences in breast cancer survival rates are influenced by a variety of factors that lead to either higher or lower outcomes. These factors include:Early Detection and ScreeningLate DiagnosisSocioeconomic and Racial DisparitiesGenetic and Biological FactorsHealthcare Access and Quality

To address complications arising from breast cancer and improve survival rates, various treatment approaches are employed. These include:

Surgical Resection: Surgically removing the tumor is often one of the first lines of treatment, particularly in early-stage breast cancer. The earlier the tumor is removed, the higher the chances of survival.

Radiotherapy: This uses high-energy radiation to kill cancer cells, often used after surgery to eliminate remaining cancerous tissues.

Adjuvant Chemotherapy: Chemotherapy administered after surgery to prevent cancer recurrence by targeting any remaining cancer cells.

Neoadjuvant Chemotherapy: Chemotherapy is given before surgery to shrink the tumor, making it easier to remove surgically.

Hormonal Therapies: These are used when breast cancer is hormone receptor-positive, helping to block hormones like estrogen that fuel cancer growth.

Monoclonal Antibodies: Targeted therapy using antibodies, such as trastuzumab (Herceptin), is particularly effective against HER2-positive breast cancers.

Immunotherapy: This approach helps boost the body’s immune system to fight cancer, showing promise in aggressive breast cancer types.

Small Molecular Inhibitors: These target specific molecular pathways involved in cancer cell growth and survival, offering a more tailored and often less toxic approach compared to traditional chemotherapy.

The selection and combination of these treatments are personalized based on the patient’s cancer type, stage, and genetic profile to improve survival outcomes and minimize complications [3]. Due to the drawbacks of the current strategy, including toxicity and side effects, the scientific community is now exploring efficient plant-based alternatives for breast cancer treatment. Furthermore, the scientific community has also focused on diminishing the limitations, including growing resistance to various drugs and toxicities linked with the existing treatment modalities. Such movements can diminish the frequency corresponding to the rising burden of complications associated with cancer across the globe as an effective alternative way. As mentioned earlier, the development of efficient and novel strategies is pivotal to managing patients with breast cancer. Hence, because of the safe nature, easy availability, and nontoxicity of predominant elements found in various medicinal plants, the recent research trends have feasibly focused on the exploitation of such plant-based elements for combating the impacts instigated by cancer, especially for breast cancer treatment. The anticancer activity of numerous medicinal plants is attributable to phytochemicals. All phytochemicals derived from plants, including the *Dimocarpus longan*, *Piper longum*, *Withania somnifera*, *Nigella sativa*, *Curcuma longa*, *Murraya koenigii*, as well as *Amora rohituka* are of significant importance in drug development. Recently, the focus of research has shifted toward formulating targeted phytochemicals that could be used to relieve the toxic effects of cancer treatment [4]. Across the globe, action is being taken to implement these measures, and this calls for a detailed evaluation of such action. This review evaluates the evidence regarding the use of plant chemicals in the prevention and treatment of breast cancers, including working mechanisms of plant chemicals, possible protective effects, dietary sources, and in vitro, animal, and clinical trial studies [5].

## 2. Breast Cancer Overview

According to the Centers for Disease Control and Prevention (CDC), breast cancer is defined as a disease in which cells grow out of control within the breast. The classification of breast cancer primarily relies on the type of cells which is turned into cancer. As per recent statistics [6], it was inferred that about two million new cases were reported in 2020, and the incidence rate has also increased over the past few years. Breast cancer can be divided into two main categories: non-invasive and invasive. Non-invasive breast cancer, such as Ductal Carcinoma In Situ (DCIS), is confined to ducts and does not spread to adjacent tissues. In contrast, invasive breast cancer spreads to surrounding connective and fatty tissues. Furthermore, breast cancer is a heterogeneous disease, and its treatment depends on the expression of surface markers like hormone receptors (HR) and HER2. The primary subtypes of breast cancer include Hormone Receptor-Positive (HR+)/HER2-Negative, which accounts for about 70% of cases and is treated with hormone therapies like tamoxifen. HER2-Positive Breast Cancer is marked by HER2 overexpression, accounts for 15–20% of cases, and is treated with HER2-targeted therapies such as trastuzumab. Lastly, Triple-Negative Breast Cancer (TNBC) lacks ER, PR, and HER2 expression and is often more aggressive, requiring chemotherapy as the main treatment, with ongoing research into alternative therapies [7,8,9]. The identification of risk factors for breast cancer is essential for effective screening and prevention. Seven key risk factors are associated with an increased likelihood of developing breast cancer: age, gender, personal and family history of breast cancer, histologic risk factors, reproductive factors, exogenous hormone use, and genetic predisposition. Notably, the risk of breast cancer rises with age, and individuals with first-degree relatives who have the disease face a 2–3 times higher risk of developing it [10].

## 3. Phytochemicals and Breast Cancer

Plants have been known as magnificent sources of phytochemical components that support the well-being of humans. Lifestyle changes escalated the rate of human diseases, especially cancer. Global statistics show the ferocity of cancer as it causes 10 million deaths worldwide [2]. As mentioned earlier, among females, breast cancer is considered the second most pervasive malignancy, leading to death [3]. Radiation therapy, chemotherapy, or surgeries are the general methods available for breast cancer treatment. However, the resistance of malignant cells still gives rise to mortality. Modern therapeutics and the latest medication can cause toxic side effects on human health and social and economic life. This condition paved a route toward traditional plant-based compounds [4] (Table 1). The need for cost-effective and safe therapeutic agents forced the scientific community to search for novel therapeutic agents of botanical origin. Since ancient times, traditional medical practitioners have been using phytochemicals as a remedy for various diseases [5]. Recent studies have proved the effect of various phytochemicals in cancer treatment [6]. Hence, the current strand involves the recognition of the anticancer potential of phytochemicals derived from various plant species. Phytochemicals, including curcumin, resveratrol, and EGCG, have effectively modulated cancer pathways such as MAPK and JAK/STAT3. These mechanisms are involved in tumor cells’ proliferation, apoptosis, and metastasis. Previous work in this field has provided a detailed review of the subject. In contrast, our review provides evidence for the contribution of these compounds to regulating these pathways through summing results from primary experimental studies.

### 3.1. What Are Phytochemicals?

Phytochemicals or secondary metabolites are the biological substances produced by plants. During its lifetime, a plant can produce primary metabolites, in particular, which are responsible for the growth of the plant body. Such components include carbohydrates, proteins, fatty acids, and other cell wall components that build plant cells [5] and help in processes such as photosynthesis, respiration, etc. [7]. Apart from that, when these plants are reconciled to adverse environmental conditions (biotic or abiotic stress conditions), the plant cells produce various secondary metabolites derived from primary metabolites, having physiological and ecological impacts. In essence, secondary metabolites make the plant competitive in its stress condition. On the other hand, secondary metabolites are produced to carry out physiological functions like symbiosis, pollination, and other significant processes. Secondary metabolites consist of complex structure and function. Based on the chemical nature and functions, secondary metabolites are mainly differentiated into the following categories: phenolic compounds, alkaloids or nitrogen-containing compounds, terpenoids, and flavonoid compounds [8].

#### 3.1.1. Phenolic Compounds

These are the most abundant secondary metabolites present in plants, derived from the phenylpropanoid pathway. Plant polyphenols are a hot topic in the scientific world as an antioxidant agent. These compounds consist of one or more hydroxyl groups with aromatic rings. The major functions of phenolics involve hostility against predators and pathogens and protection from ultraviolet radiation [9].

#### 3.1.2. Alkaloids

Alkaloids are principally found in plants and are made up of hydrogen, carbon, nitrogen, and oxygen. Depending on the chemical structure, elements, and sources, the alkaloids are classified into various classes. Compounds such as caffeine and nicotine fall under the group of alkaloids [7]. Classification based on the biogenesis of alkaloids yields three types: true alkaloids, protoalkaloids, and pseudoalkaloids. The first two types were produced from amino acids, while the later types were not formed from these elements. The classification based on the structure differentiated the alkaloids into various categories enlisted in Table 2.

#### 3.1.3. Terpenoids

Terpenoids or isoprenoids are the modified class of terpenes along with functional groups. These groups of chemicals come under myriad groups of chemicals with basic functions such as growth, development, and protection from stress conditions [10]. Moreover, humans use terpenoids derived from plants in the pharmaceutical, food, and chemical industries.

#### 3.1.4. Flavonoids

Flavonoids are compounds derived from green plants and are highly recommended for their medicinal and nutraceutical properties [12]. Various potential, including the antioxidant, anti-carcinogenic, and anti-inflammatory properties of flavonoids, are getting attracted worldwide.

#### 3.1.5. Glycosides

The chemical elements belonging to the class of Glycosides usually consist of one or more sugars joined with complex molecules or a phenol or an alcohol. The non-sugar moiety of chemical elements is usually termed as a genin. The potential of such elements to combat the drastic impacts instigated by cancer has been previously mentioned [13] in many instances (Table 3).

### 3.2. Potential Benefits of Phytochemicals

There is adequate research available on the potential of phytochemicals to maintain human health. Nowadays, most of the research is based on the ineffectiveness of contemporary treatments and their negative impacts [19]. A study by [20] revealed that most medicinal plants contain phytochemicals with anticancer properties. Phytochemicals are an auspicious element for the treatment of breast cancers, irrespective of estrogen dependency [3]. A multi-targeted mechanism of action by phytochemicals is responsible for the above-mentioned anticancer activity in a majority of conditions, and advanced phytochemical delivery to the affected cells also improves the state in many instances. Besides, these elements attack the signaling pathway of cancer cells. Polymeric nanoparticles are one of them, and the phytochemicals encompassed by polymeric nanoparticles are used for drug delivery to malignant breast cells. Moreover, plants like *Curcuma longa*, *Nigella sativa*, *Cantharanthus roses*, and *Rauvolfia serpentine* have the potential to fight against breast cancer cells.

Various phytochemicals are available to use as a Selective estrogen receptor modulator (SERM) that acts as a barrier to Estrogen receptor (ER) and hormone binding process, which may lead to breast cancer. Furthermore, scientific studies have proved the potential of various phytochemicals, such as lignans and isoflavones, to cure breast cancer [3]. Similarly, a study by [21] demonstrated the effect of polyphenolic catechin present in green tea to alleviate the growth of breast cancer cells. Israel et al. concluded their study by stating the antiproliferative activity of flavonoids in suppressing breast cancer [3]. The findings of such studies also directed the authors to search for the possible mechanism behind the action.

### 3.3. Mechanisms of Action

The molecular mechanisms of phytochemicals, as shown in Figure 1, with breast cancer cells, are quite intricate and specific [22,23,24,25]. In the case of breast cancer, conventional treatment methods confer failures in patients in many instances, and it demands a modified drug delivery apparatus. Moreover, the various inferences concerning the possible action of phytochemical elements toward the cancerous cells have been given in Figure 1 and Figure 2, which also justify the role of the following signaling pathways: Akt/PI3K/mTOR signaling pathway, MAPK signaling pathway, and JAK/STAT signaling pathway [26,27,28]. As given in Figure 1, a lot of molecular mechanisms are allied with phytochemical administration.

Furthermore, Figure 2 depicts the various signal pathways that have a prominent role in the regulation.

Integrin, in conjunction with HER2, is important in the escalation of breast cancer because, as mentioned before, it fosters tumor cell adhesion, proliferation, and metastasis. Tyrosine Kinase Receptor HER2 forms a signaling complex with integrins, which activates downstream pathways like focal adhesion kinase (FAK) and PI3K/Akt [29].

Resveratrol and quercetin are phytochemicals that appear to block the HER2-integrin signaling axis. It has been established that resveratrol prevents HER2 phosphorylation and subsequently blocks integrin β1 signaling, which inhibits migration and cell invasion by inhibiting the downstream FAK activation [30]. Quercetin encourages proteasomal degradation of the second HER2, modifying integrin-mediated adhesion and migration. Quercetin aids in HER2 degradation through modulation of the ubiquitination pathways alongside resveratrol, which interferes with phosphorylation sites on HER2, greatly reducing the interaction that HER2 has with Integrins [31].

The pathway is crucial for cell proliferation, differentiation, and cell survival. In breast carcinoma, the aberration in this pathway is instrumental to tumor formation and the resultant evasion of apoptosis [32].

Phytochemicals, including curcumin and EGCG, can interact with and modulate the MAPK pathway. The action of curcumin prevents the phosphorylation of MEK1/2, which in turn restricts the activation of ERK and the subsequent regulation of various transcription factors. Furthermore, EGCG inhibits crucial MAPK components such as c-Jun N terminal kinase (JNK) and ERK, therefore aiding in reducing inflammation and overall tumor development [33,34].

Curcumin interacts with MEK1/2 directly by locking into their kinase domains. This action prevents phosphorylation from occurring. On the other hand, EGCG does not interact specifically with MAPK signaling; rather, it indirectly inhibits receptor tyrosine kinases upstream, which results in weaker activation of the cascade downstream [35].

As reviewed earlier, phytochemicals are substances with wide properties against breast cancer cells. Due to the negative impacts of current breast cancer treatment methods, phytochemicals are a promising alternative for breast cancer therapy. This comprehension gave birth to various advanced technologies using phytochemicals, but still, they are not available to mankind. The various studies discussing the major role of phytochemicals in cancer research in context with the pharmacological and therapeutic benefits have been given in Table 3. The cancer target, therapeutic effect, and the type of phytochemical constituent in the study are evident in Table 4.

## 4. Phytochemicals Found in Foods

The knowledge of the relation between food and diet made us realize the concept of ’food as medicine’. Phytochemicals are rich natural sources with non-toxic, safe, cost-effective, and biomedical properties. Various food crops have already been explored for phytochemicals with antioxidant, anticancer, and anti-inflammatory properties. Other properties such as emulsifying, stabilizing, antioxidant, and aroma include these phytochemicals in the pharmaceutical, cosmetics, and food industries [44]. Fruits, vegetables, and roots are the major sources of phytochemicals (Figure 3). The phytochemicals are directly available from natural resources such as *Casatanea sativa* for polyphenols, *Solanum melongena* for glycoalkaloids, and papaya fruit for phenolic compounds [45].

### 4.1. Cruciferous Vegetables

Cruciferous vegetables are popular in the global diet. Most of the studies suggest the presence of phytochemicals glucosinolates and sulforaphane in cruciferous vegetables, including broccoli and cauliflower [46]. Previous studies have evaluated the anticancer effects of glucosinolates and sulforaphane through in vitro assays. These studies frequently involve the use of cultured breast cancer cell lines to assess cell viability, proliferation, and apoptosis. For example, in vitro studies have demonstrated that sulforaphane can inhibit cell proliferation and induce apoptosis in various breast cancer cell lines by modulating pathways such as the PI3K/Akt and NF-κB signaling pathways, which are crucial for cell survival and inflammation [47].

Additionally, some in vivo studies have been conducted to verify these findings in animal models of breast cancer. These studies generally report favorable outcomes, such as reduced tumor growth and metastasis, upon dietary or direct administration of glucosinolates or sulforaphane extracts from cruciferous vegetables. For instance, recent research using murine models has highlighted sulforaphane’s potential to not only inhibit primary tumor growth but also suppress metastasis, enhancing its relevance as a dietary approach to breast cancer management [48].

These findings support the rationale for further exploration of these phytochemicals, as understanding their precise mechanisms could aid in developing new therapeutic strategies, particularly given their accessibility through diet. The study seeks to build on these in vitro and in vivo outcomes, examining the mechanisms by which glucosinolates and sulforaphane interact with breast cancer pathways. Focusing on these pathways may also provide insights into the potential clinical translation of these compounds for breast cancer prevention or adjunct treatment.

### 4.2. Berries and Citrus Fruits

Citrus fruits are consumed globally due to their taste, aroma, and health benefits. Scientific research has proven that citrus contains phytochemicals such as carotenoids, flavonoids, various essential oils, and limonoids [49]. Similarly, berries are a rich source of phytochemicals like polyphenols and alkaloids [50]. Diab et al. witnessed the exploitation of citrus fruits and their parts for breast cancer research [47].

### 4.3. Green Tea and Other Beverages

Tea and green tea are the most popular drinks with a rich source of antioxidants. Catechins, flavonols, polyphenols, and quercetin are some of the known phytochemicals present in green tea extract [48]. The presence of various compounds from these sources produced significant effects in combating breast cancer [51].

### 4.4. Herbs and Spices

Herbs and spices have been consumed for decades and are important in flavoring, medicinal purposes, breast cancer treatment [52], and cosmetics preparation [53]. Ref. [54] studied the presence of phytoconstituents in plants like basil, ginger, garlic, lemongrass, parsley, oregano, chili, and Italian herbs. The study concludes with evidence of the presence of phenolic compounds in all the species. Flavonoids are the content that provides color to the herbs [44]. Previous studies also [45] explained the phenolic content present in *Oenanthe javanica*, a perennial herb.

### 4.5. From Marine Sources

The dates back to 1940 have principally witnessed the discovery of various chemical elements with potent pharmacological benefits. The various efforts during the mentioned period verified the potent use of marine organisms as a prominent natural source to combat life-threatening diseases, including cancer. In this regard, ref. [55] enlisted and analyzed the pharmacological benefits of the following compounds with special reference to anticancer potential: Cytarabine, ARA-C, Brentuximab vedotin, Hemiasterlin, Bryostatin 1, Taltobulin, Pseudopterosins, Plitidepsin, and Trabectedin. In addition, ref. [55] also reviewed the possibilities of exploiting the following for anticancer potential: marine algae (brominated phenols, kainic acids, sterols, phenazine derivatives, nitrogen-containing heterocyclics, prostaglandins, guanidine derivatives, amines, and sulfated polysaccharides), corals reefs (dolastatin, nitrogenous diterpene, and sterols), marine herbs (alkaloids, polyphenols, and polysaccharide), sponges (renieramycin M and heteronemin), sea weeds (dexcyanidanol, trihydroxybenzoic acid and catechuic acid), marine bacteria (riamycin, mitomycin C, daunorubicin, lapachol and streptonigrin), and marine ascidiaceans [halocynthiaxanthin, brominated 3-(2-aminopyrimidine)-indoles, and fucoxanthinol].

**Figure 3 cimb-47-00030-f003:**
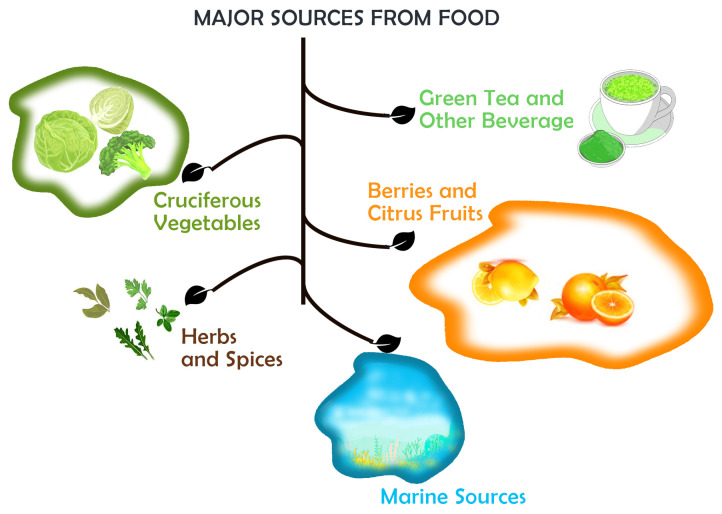
Major sources of phytochemicals from food.

## 5. Studies on Phytochemicals and Breast Cancer

The former strands of the study have clearly mentioned that breast cancer is one of the major causes of global cancer mortality in women. Mortality rate increases due to the recurrence of cancer cells, resistance to treatment methods, and gloomy prognoses of breast cancer. Even in this era with modified treatment methods, we fail to come out from the conventional methods of breast cancer treatments such as surgery, chemotherapy, and radiation therapy [56]. Hurdles associated with regular treatment methods demand the discovery of innovative, cost-effective, non-toxic, and novel therapeutic agents. Recent investigations suggest phytochemicals derived from plants as an alternative to breast cancer treatment [57]. Phytochemicals refer to the bioactive compounds derived from plants with complex chemical structures. Due to this potential, many of the food crops with various phytochemicals have been incorporated into our diet in ancient times. More than 5000 phytochemicals have been recorded from various fruits, vegetables, and other food crops [58]. Scientific studies have proved the health benefits of phytochemicals as antioxidant, antibacterial, anticancer, and anti-inflammatory. Moreover, the effect of phytochemicals on fighting breast cancer cells is an appreciable topic to discuss.

### 5.1. In Vitro Studies

Various investigations showed beyond doubt that phytochemicals are capable of preventing changes in estrogen-dependence as well as independent proliferation of breast cancer cells. Research is frequently conducted to discover the inherent properties of phytochemicals derived from various plant species. After the introduction of phytochemicals, the antioxidants work to prevent the development of cancer cells. The termination of active tumor cells by phytochemicals is discussing the possibility of incorporating the secondary metabolites in the drug delivery system. Various in vitro and in vivo studies have been conducted to check the anticancer potential of phytochemicals [59]. Silibinin is a form of flavanone compound that can be extracted from various medicinal plants [60]. The anticancer potential of silibinin was already documented. Ref. [61] studied the capability of silibinin to treat human breast carcinoma cells. They concluded that the phytochemical application decreased the spreading of cancer cells.

Cell line studies conducted by [62] revealed the properties of naringenin and silibinin, the phytochemicals that prevent the effect of CYP 19, which is a gene that plays an important role in the production of estrogen. Ref. [63] conducted an in vitro study on the phytochemical potential and commented on the combined effect of phytochemicals against malignant cells. Moreover, in vitro studies by [64,65] demonstrated flavonoids and their antiproliferative effect on carcinogenic cells. The inhibition properties of naringenin compounds on cancer cells were also discussed in this study. Similarly, in vitro studies by [66] have explained curcumin, which can control the expression of FEN1, a protein-coding gene. As a result, the susceptibility of breast cancer cells to cisplatin was increased. In their research, Jia et al. (2014) examined how curcumin impacted two breast cancer cell lines—MCF-7 and MDA-MB-231. The research revealed that the response of these cell lines to curcumin treatment was different and was dependent on the modulation of the PI3K/Akt-SKP2-Cip/Kips pathway. To elaborate, in MDA-MB-231 cells, curcumin exposure is associated with decreased cell viability and clonogenic survival in correlation with the modulation of the above signaling pathways. While quantifiable levels of ERK1/2 were not accessible within the research, it was outlined how curcumin has a profound effect on vital signaling networks associated with cell growth and survival [67].

A study conducted by Fu H et al. aimed to determine the mechanisms of curcumin action on two gastric cancer cell lines, SGC-7901 and BGC-823, using MTT, flow cytometry, TEM, and Western Blot assays. The expressions of P53 and P21 were upregulated, while PI3K, p-Akt, and p-mTOR were downregulated. Curcumin facilitated apoptosis, autophagy, and P53 pathway signaling together with cell proliferation inhibition curcumin, as confirmed by the western blot analysis [68].

### 5.2. Animal Studies

The anticancer properties of plant secondary metabolites have been analyzed by various researchers through in vivo studies. Ref. [69] revealed a clear view of the effect of reactive oxygen species (ROS) in animals induced with breast cancer. A similar study explains the use of nanoparticles for drug delivery along with the antioxidant agent. Curcumin is one of the efficient phytochemicals with various biomedical properties. In vivo studies are evident for the anticancer properties of curcumin in a dose ranging from 300 mg to 3500 mg/kg body weight [70]. Homoharringtonine is another phytochemical present in various tree species. Generally, this compound controls the proliferation and growth of cancerous cells and malignant cells. This was the finding repeated by [71]. They described the anticancer property through an in vivo study, and the result after 5 days of treatment was the inhibition of MCF-7 breast cancer cells. In short, homoharringtonine can suppress the proliferation of breast cancer cells and apoptosis. In addition, in vivo studies showed that 30mg/kg of EGCG for two weeks reduced the count of hepatic metastatic lesions and prevented tumor proliferation while promoting apoptosis in the tissues. Moreover, treated tumors were considerably less vascularized than those in the controls [72].

A finding from the RNA-seq examination analysis revealed that the genes under the influence due to RSV treatment were concerned with apoptosis and the p53 signaling pathway. Furthermore, POLD1 was noted to be the major mediator in the apoptosis of MDA-MB-231 cells as caused by RSV. The full-length PARP1, PUMA, and BCL-2 expression expressed that the cells were treated with RSV and that the levels of PCNA were noticeably decreased while the levels of Cleaved-PARP1 and Cleaved-Caspase3 rose sharply [73].

### 5.3. Human Studies

Experimentation of new therapies under the supervision of scientists is important for the development of effective drugs [74]. It is difficult to conduct human trials to develop a particular medicine. Major social, cultural, and economic issues will arise for trial in the human body. Moreover, the trials take place after the drug combinations are confirmed with animal models. Preclinical data suggest that resveratrol can diminish HER2-mediated integrin signaling, which may result in decreased metastasis in breast cancer models that exhibit overexpression of HER2 receptors. In preclinical models, both EGCG and curcumin exhibited dramatic efficacy as they inhibited the action of MAPK, thereby making them potential adjunct therapies for resistant forms of breast cancer [75].

### 5.4. Molecular Docking

In recent years, we have also witnessed the use of advanced approaches, such as molecular docking against breast cancer. A study by [76] explored the possibilities of implementing molecular dynamic and molecular docking of various products of botanical origin targeting breast cancer treatment. They used the following ligands: Hecogenin acetate, Podototarin, Hesperidin, and Theaflavin. The toxic potential of such elements towards the cancer cell lines—precisely, the capacity to induce toxicity against cancer cells (breast, colon, and lung liver)—make them effective candidates for future research.

## 6. Phytochemicals as Potential Preventive Agents

Phytochemicals are plant-based sources with tremendous protective ability against disease-causing agents. Properties like antioxidant and anticancer activities show the potential of phytoconstituents as a source of new medication. The mechanism of phytochemical actions is complex and unique. Most phytochemicals exhibit distinctive activities as preventive agents. They help to control the molecular pathways that lead to cancer. In the case of phytochemicals with anticancer activity, flavonoids, alkaloids, terpenoids, and phytotoxins have separate modes of action.

### 6.1. Role in Breast Cancer Prevention

As discussed in earlier parts of the study, many of the plant-derived products are primarily used for the treatment and management of deadly diseases. The available plant-based products are either entire plants or specific parts or a particular form like capsule, powder, or liquid forms. Such products can be consumed as tablets, food, or gels to spread over the skin and, in certain instances, can be used in bathing water. However, appropriate forethought should be held during the use of such plant-based products in patients since there exists a chance of emerging adverse to the health of patients [77]. The previous studies on this aspect revealed that the combination of herbal products with conventional anticancer medicines readily diminished the major side effects instigated by synthetic drug-based chemotherapies [78]. Among the various strategies implemented in this regard, dietetic phytochemicals are extensively used in the chemoprevention process. Because of the following reasons, this strategy would be getting much more significance than other ones: cost-effectiveness, the complex structure of chemicals, reduced toxic potential, easy accessibility, and various biological effects. The phytocompounds are also reported to have the ability to alter multiple signaling pathways and develop a defensive role in cancer chemoprevention. The dietary chemopreventive phytochemical constituents not only played a significant role in preventing the drastic impacts instigated by cancer, but they were employed as a primary competitor for the development of efficient chemotherapeutic drugs for breast cancer. Table 5 lists the major compounds that are primarily used for the treatment of breast cancer.

### 6.2. Effects on Tumor Growth and Progression

Phytochemicals are highly recommended as a cancer treatment agent due to their potential in the prevention of cancer cell development. The findings from various previous articles on this direction in a concentration-dependent manner have been given in Table 6. Nowadays, physiologically potential phytochemicals are widely used for cancer studies to create an alternative remedy for cancer cells. Regulation of molecular pathways, reduction in oxidative stress, inactivation of cancer cells, regulation of cellular proliferation, and control of the immune system are the modes of action of phytochemicals within an infected body. Phytochemicals can regulate the mode of development of cancer cells. Action, along with signaling pathways, can support the control of the initiation of malignant cells and the promotion and progression of tumor cells by applying the anticancer and antioxidant properties of secondary metabolites [90].

Inhibition of tumor cells is occurring as a result of bringing about apoptosis. The mechanisms happening in the formation of tumors are complex. It undergoes epithelial-mesenchymal transformation. The scientific community widely accepts studies on the anticancer activity of various plants. Phytochemicals can prevent the action of genotoxic carcinogens from forming adducts with DNA. This may happen by the inhibition of carcinogens or by detoxification [104].

### 6.3. Potential Side Effects and Interactions

Along with the tremendous positive sides of phytochemicals, there are also some adverse effects for them. Resveratrol is a group of polyphenols, one of the phytochemicals that are used to manufacture chemical compounds for cancer treatment. These compounds are widely present in crops like grapes, cocoa, and berries [105]. Many of the research findings supported the advantages of resveratrol as below: In vitro and in vivo studies have proved the potential of resveratrol to act as an antiproliferative, plant antibiotic agent with antitumor, antimutagen, antioxidant, and anti-inflammatory activities. Initiation, promotion, and progression are the various stages of tumor development. During the induction of resveratrol, it reacts to the metabolizing enzymes to prevent the initiation of tumor activity. It can be used for the treatment of breast cancer cells due to its action against MCF-7 cancer cells. The following studies have reported the potential side effects [106,107,108].

More than that, it is highly recommended for its antipromotion activity and antiprogression activity. At the same time, various researchers are giving data on the negative impacts of resveratrol. The result of an in vivo study on the toxicity of resveratrol explains the renal toxicity in rats. It was observed that the low concentration of some of the phytochemicals that are used for cancer treatment has been reported to have toxic effects in certain instances. Precisely, the complications allied with gastrointestinal anomalies, liver, hormones, allergic reactions, neurons, and blood are witnessed in previous investigations.

### 6.4. Challenges and Limitations

Phytochemicals promote HER2 and MAPK pathways in a clinical context; however, there are some drawbacks, such as low bioavailability and variability in phytochemical extract composition. Utilizing nanoparticles as a delivery system and standardized extraction protocols are some of the methods to achieve these goals.

## 7. Phytochemicals as Adjunctive Treatment

Phytochemicals come under various chemical groups. Major phytochemicals, including alkaloids, flavonoids, terpenoids, coumarins, polysaccharides, carotenoids, and indole, are highly studied to analyze their biomedical potential. Their antioxidant, antimicrobial, anticancer, and anti-inflammatory activities are the reasons for scientific concern. The direct action of most of the phytochemicals has not been explored much, and many of the compounds are used for drug delivery. Future studies in humans are encouraged to explore the potential of various phytochemicals, as plant-derived compounds have demonstrated promising health benefits through in vitro and in vivo models. The direct effects of most phytochemicals remain largely unexplored, with many compounds currently applied primarily in drug delivery systems. However, there is limited research on their optimal concentrations and use in supportive therapies. Phytochemicals present in fruits, vegetables, and herbs have the potential to be used clinically owing to their antioxidant, anti-inflammatory, and anticancer activities. However, extensive clinical studies are necessary to determine the safety, effectiveness, and concentration required for human administration. This must be attained before phytochemicals can be applied to clinical practice and included in dietary guidelines [109].

### 7.1. Complementary Approaches

Cancer is a deadly disease, and the drawbacks of conventional treatment methods have made the scientific world think about alternative treatment methods. Phytochemicals are the most suitable remedy for the betterment of this condition [110]. Phytochemicals are the chemical compounds that are present in fruits, vegetables, legumes, grains, and spices. These constituents possess highly effective substances for the control of signaling pathways. Most cancer forms are occurring due to environmental conditions and food habits. Reactive oxygen species formed due to stress conditions, and food generates oxidative stress and DNA damage [110]. When the intake of free radicals through a proper diet and the detoxification mechanism come in balance, then the prevention of cancer cells is possible. Compared with conventional methods, food including plants with rich phytochemicals is highly effective in cancer treatment. It is considered an economical, safe, and biological method with fewer risk factors.

### 7.2. Synergistic Effects with Conventional Treatments

Synergy is the effect of a compound while combining another substance to improve the expected result. Here, the effect of phytochemicals improves when combined with conventional treatment methods. Any of the phytochemical constituents combined with a conventional drug or treatment method for drug delivery and the mode of action of the drug-compound combination will be monitored [111]. Damaging agents are highly susceptible to breakdown during the application of a multi-drug. This combination can affect various target molecules such as proteins, cellular metabolites, nucleic acids, enzymes, receptors, etc. This can effectively improve the mechanism of prevention of cancer cells. The following studies have prominently exploited the possibilities of implementing Synergistic approaches for better outcomes [112,113,114,115,116].

### 7.3. Challenges and Limitations

Phytochemicals are known to have low bioavailability because they are rapidly metabolized, have low solubility, and have low absorption. For example, curcumin is a potential anticancer drug, but due to rapid first-pass metabolism and elimination, it has low systemic bioavailability when dosed orally [117]. Various nanotechnology-based delivery systems, such as liposomes, nanoemulsions, and nanoparticles, have emerged to improve their solubility and pharmacokinetic parameters to address this issue. As indicated through preclinical and early development phase clinical studies, these strategies improve delivery, including outstanding targeted delivery, stability, and high solubility [118].

Plant-derived phytochemicals vary in composition due to differences in species, growing conditions, and extraction methods, making standardization challenging. Such variability impacts reproducibility and efficacy in clinical settings [119]. Standardization of plant extracts using validated analytical methods, such as high-performance liquid chromatography (HPLC), ensures consistent composition and potency. Regulatory guidelines for botanical drugs can also provide frameworks for standardization [120].

Phytochemicals are relatively non-toxic, albeit administering them at dosage levels higher than permissible or combined with chemotherapeutic agents can have some adverse effects [121]. For example, resveratrol has been linked in some studies to gastrointestinal distress and renal toxicity, which is still debatable as the agent is considered to be most beneficial [122].

Furthermore, drug–phytochemical interactions with anticoagulants and other polyphenols can lead to bleeding [123]. Compiling detailed databases of drug–phytochemical interactions and employing them alongside preclinical toxicological investigations of products can assist clinicians in reducing the risk of such events [75].

## 8. Future Directions and Research Opportunities

Globally, cancer is considered one of the major causes of death among people, and the mortality rate demands improved technologies and novel substances for treatment. Dangerous side effects of conventional cancer treatment methods are seeking plant-based, safer, and economically beneficial methods for the therapy. The future of cancer treatment belongs to phytochemicals derived from plants. Cancer regulation is based on the prevention of malignant cells, delaying cell growth, and curing the cells. Many of the phytochemicals, such as flavonoids, carotenoids, alkaloids, polyphenols, etc., contain antioxidant, antimicrobial, anticancer, and anti-inflammatory properties. Extensive examinations of the biomedical properties of these herbal compounds have been conducted, but their potential has not been explored completely. The future of this area is vast and quite interesting. Our flora is uncountable, and the chemical constituents that occur in those plants are also unpredictable [124].

### 8.1. Identifying Novel Phytochemicals

The complex structure of phytochemicals groups them into various classes. Based on their activity, phytochemicals are mainly identified as compounds such as vinca alkaloids, taxane diterpenoids, camptothecin derivatives, and epipodophyllotoxin [74]. Apart from these phytochemical classes, other plant-derived anticancer agents from different classes, such as combretastatins, homoharringtonine (omacetaxine mepesuccinate, cephalotaxine alkaloid), and ingenolmebutate are also used as anticancer agents. Poor aqueous solubility and significant toxic side effects still remain major concerns, and therefore, the current focus of research is on eradicating the impact of these factors. In this context, several analogs and prodrugs have been synthesized, and methods have been devised to enhance aqueous solubility and tumor specificity.

### 8.2. Optimizing Dosages and Delivery Methods

Most of the natural compounds have various biomedical activities but are struggling due to low availability and selectivity, which may interrupt the activity of the drug. A study by [125] explained the bioavailability of various secondary metabolites such as curcumin, resveratrol, flavonoids, berberine, and camptothecins. The methods used for the analysis of the dose were liposomes, nanoparticles, nanoemulsions, and films. The study concludes the dose dependency of phytochemicals in drug delivery methods.

### 8.3. Personalized Medicine Approaches

The influence of plant-based substances is high in target-based therapies [126]. The multi-targeting capability of plant-based phytochemicals is highly appreciable. Decisions on personalized medicine are based on the right drug selection, appropriate dose selection, duration of application, and also a correct combination. Genomic profiling as a potential strategy for personalized Phytochemical treatments.

With advancements in genomic profiling, for instance, the BRCA1/2, HER2, and PIK3CA genes, usually evaluated in breast cancer patients, are now easy to identify. Genomic profiling includes sequencing the patient’s DNA to detect genetic variations or mutations that may contribute to the development of certain types of cancer or affect treatment outcomes. For breast cancer, for example, mutations in the BRCA1, BRCA2, PIK3CA, or TP53 genes may play a role in the progression of tumors and their treatment [127]. This information allows phytochemicals to be selected or altered to affect a specific genetic pathway.

For instance, curcumin, which comes from turmeric, blocks the PI3K/Akt/mTOR signaling pathway frequently activated in breast cancer containing PIK3CA mutations [128]. Isoflavone, a soy extract, also suppresses ER-mediated signaling and may be useful for treating ER-positive breast cancer [129].

#### 8.3.1. Importance of Biomarkers in Tailored Phytochemical Selection

Breast cancer patients can benefit from targeted therapies based on their biomarker profile (ER, PR, and HER2). Phytochemicals may also be possible as targeted therapies based on these biomarkers [130].

According to research, resveratrol may act as a HER2 target for treating human cancers [30]. The isoflavones daidzein and genistein can exhibit estrogenic activities, which may be useful for treating ER-positive cancers [30]. However, they must be used with care in patients with hormone-dependent cancers to prevent negative outcomes.

Furthermore, ctDNA and micro-inhibiting RNAs can serve as new molecular markers for real-time tracking of treatment effectiveness [131]. Adding phytochemicals to these tools can also enhance treatment strategies.

#### 8.3.2. Future Perspectives on Personalized Phytochemical Therapies

Advancements in artificial intelligence (AI) and bioinformatics can enable the formulation of predictive models that correlate phytochemical profiles to patients’ genomic and biomarker profiles. Such approaches could vastly enhance the use of phytochemicals in precision oncology [132].

AI can analyze genomic and protein data to forecast how cancer-causing markers engage with a certain bioactive compound. Furthermore, CRISPR technologies could be used to alter the structure of the biomarker-targeting phytochemical to enhance its binding capacity [133].

Molecular docking simulations and genomic profiling have been utilized to search for breast cancer-specific plant molecules, especially those targeting components associated with JAK-STAT signaling pathways [133].

### 8.4. Integrating Clinical Trials

Applying phytochemicals into practice should be based on strong evidence that the clinical trials provide. Several trials have assessed the effectiveness, safety, and mechanisms of action of phytochemicals in cancer therapy, particularly in breast cancer:

A Clinical Trial of Curcumin Co-therapy with Docetaxel in Patients with Advanced Breast Cancer Programme: A Phase II clinical trial showcased its safety and tolerance when combined with docetaxel in advanced breast cancer patients. This trial demonstrated curcumin’s preclinical ability to augment chemotherapy through targeting cancer cell signaling pathways such as NF-κB and PI3K/Akt [133].

*Green Tea Polyphenols (EGCG):* Other studies have been conducted, including NCT00516243, which investigates the effects of epigallocatechin gallate (EGCG) on breast cancer. The results demonstrated substantial efficacy against cancer in areas such as tumor cell growth and angiogenesis [134].

*Resveratrol and HER2-positive Breast Cancer:* Research is ongoing that aims to use resveratrol in HER2-positive breast cancer. Resveratrol converges on downregulating HER2 signaling pathways and stimulating apoptosis in tumor cells [135].


*Challenges in Translating Phytochemicals to Clinical Practice*


Even with promising phytochemicals, there are still several limitations that must be worked on to make them clinically useful:

*Lower Bioavailability:* Certain phytochemicals, such as curcumin and resveratrol, have low systemic uptake, increased metabolism, and low half-lives, negatively impacting the range of treatment options. For instance, curcumin alone is said to have less than 1% bioavailability when taken orally [135]. Ground-breaking drug delivery methods like liposomes, nanoparticles, and micelles are being made to maintain a constant therapeutic level [135].

*Variation in Composition of Plant Extract:* The type of extraction technique used, coupled with the variety of plants and environmental factors such as temperature, tremendously impacts the concentration and strength of phytochemical compositions [136]. Therefore, standardization protocols and regulatory frameworks issued by the U.S. FDA are crucial for maintaining the consistency and reproducibility of the product.

*Adverse Effects and Interactions:* Resveratrol is among certain phytochemicals that are said to have negative side effects, such as renal and gastrointestinal toxicity, especially in large doses. The use of polyphenol-rich compounds and anticoagulants together can lead to severe bleeding complications. Thus, a way to reduce these effects is through preclinical toxicology evaluation, and patient databases about drug–phytochemical interaction methods are available for patient use [137].

#### 8.4.1. Strategies to Address Challenges

To address these challenges and put the findings into reality, several approaches can be used:Pharmacokinetic Optimization

Delivery systems based on nanotechnology, including polymeric nanoparticles and nanoemulsions, could enhance the solubility, stability, and targeted delivery of low-bioavailable phytonutrients such as curcumin and EGCG.

2.Standardization of Phytochemical Products

Validated analytical methods such as high-performance liquid chromatography (HPLC) should be used to avoid inconsistencies in the constitution of the phytochemical products. Regulations such as those of the EMA and FDA offer quality control standards [137].

3.Adaptive Clinical Trial Designs

Adaptive trial designs permit alteration as new information is available, addressing variation in response to the phytochemical, especially when multiple variables are involved. Basket trials, for instance, could be used to target several cancer subtypes that have the same biological targets [138].

4.Integration of Genomic and Biomarker Data

Genomic profiling and biomarker profiling can be used to better select patients for phytochemical therapies and avoid the one-drug-fits-all strategy [138]. For example, when a stronger potential for growth like EGCG is to be used in hormone receptor-positive patients, some specific biomarker profiles may best suit this method of therapy.

#### 8.4.2. Future Directions

We recommend that further investigations aim to conduct multicentre and longitudinal studies to evaluate the effectiveness and safety of phytochemical therapies. The information obtained from the collaboration of academic and industrial Centers with regulatory bodies will help translate the data from bench to bedside. It would also be important to formulate general policies on how to modify standard oncological treatment using phytochemicals.

## 9. Conclusions

The study critically reviews the potential of phytochemicals in breast cancer treatment, highlighting their pharmacological properties, such as antioxidant, anticancer, anti-inflammatory, and antimicrobial activities. These phytochemicals, derived from plants (e.g., catechins, piperlongumine, lycopene, isoflavone, cucurbitacin, and BPEITC), show significant promise in targeting breast cancer through mechanisms like cell cycle arrest, apoptosis, and inhibition of signaling pathways (e.g., JAK-STAT3, HER2-integrin, MAPK). Despite their advantages, challenges, such as potential side effects and toxicity, remain.

Future research should focus on clinical trials, optimizing delivery systems (e.g., nanoparticles, liposomes), and integrating phytochemicals with conventional therapies to enhance efficacy. Combining genomic profiling and biomarker discovery with phytochemical therapies may individualize treatments, maximize therapeutic outcomes, minimize toxicity, and provide cost-effective alternatives, especially for economically disadvantaged populations. Emphasis is placed on understanding mechanisms of action, dosing strategies, and dietary inclusion of phytocompounds to improve breast cancer management.

## Figures and Tables

**Figure 1 cimb-47-00030-f001:**
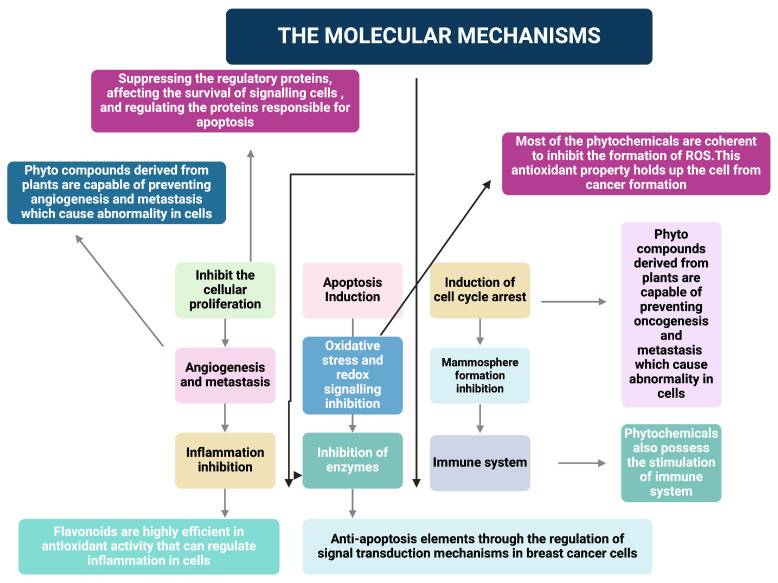
Molecular mechanisms of phytochemicals in cancer treatment. It demonstrates their roles in cellular proliferation, apoptosis, and cell cycle. Phytochemicals prevent angiogenesis, metastasis, oxidative stress, and inflammation and regulate redox signaling. They inhibit enzymes, modulate the mammosphere formation, and enhance immune system activity. The anti-apoptotic modulation and targeting of breast cancer cells and preventing the rate of perfusion look positive due to the antioxidative nature of the flavonoids.

**Figure 2 cimb-47-00030-f002:**
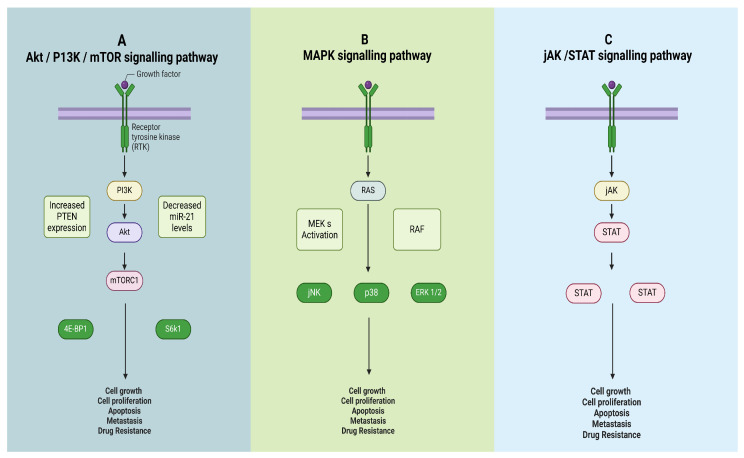
Key signaling pathways involved in cancer progression and drug resistance: (**A**) Akt/PI3K/mTOR Pathway: Growth factors activate receptor tyrosine kinases (RTKs), stimulating PI3K and Akt, leading to mTORC1 activation. This regulates cell growth, proliferation, apoptosis, metastasis, and drug resistance by modulating PTEN expression, miR-21 levels, 4E-BP1, and S6k1. (**B**) MAPK Pathway: RTK activation triggers RAS, leading to RAF and MEK activation. Downstream effectors (JNK, p38, ERK 1/2) influence similar cellular processes. (**C**) JAK/STAT Pathway: Cytokines stimulate JAK, phosphorylating STATs, which translocate to the nucleus, driving genes involved in growth and drug resistance.

**Table 1 cimb-47-00030-t001:** Phytochemicals and biomedical activities.

Phytochemicals	Plant Source	Biomedical Activity	Reference
Polyphenols			
Flavanols	Grapes, green and black tea	Antioxidant Anti-inflammatory	[11]
Flavanones	Citrus fruits, grapes	Anti-inflammatory Anti-allergic	
Flavones	Pea, watermelon, pepper	Anticancer Antioxidant	
Anthocyanidins	Cereals, legumes	Anti-inflammatory Antioxidant	
Carotenoids			
α-carotene	Banana, avocado, mango, pumpkin	Anticancer Eye Health	
Lutein	Spinach, broccoli, pepper, nuts, dates	Improves immunity Hepatoprotective	
β-carotene	Spinach, grapes, carrot, pepper, mango, lettuce	Improves immunity Hepatoprotective	
Xanthophylls	Pepper, mushroom, pumpkin	Improves blood flow Antioxidant	
Fucoxanthin	Seaweeds, microalgae	Antioxidant Anticancer Antimicrobial Antiobesity Anti-inflammatory	
Saponins			
Oleanane	Almond, black gram	Hypolipidemic Antimicrobial	
Polysaccharides			
Fucoidan	seaweeds	Antioxidant Antimicrobial	
Amylopectin	Rice, corn, potato	Improves gut microbes	
Fibers	Green leafy vegetables, fruits	Prevent cardiovascular diseases	
Terpenoids	Algae, mushrooms, lichens	Antiviral Anti-inflammatory Anti-allergic	
phytoestrogen	Berries, grapes, peanuts, peas, cereals	Protection from cardiovascular diseases Anticancer Antidiabetic	

**Table 2 cimb-47-00030-t002:** Major alkaloid compounds and origin.

Sl. No.	Group	Major Plant Source/Family/Species	Chemical Nature/Nucleus	Structure of the Nucleus
1.	Tropane alkaloid	Solanaceae	Tropane (C_4_N skeleton) nucleus	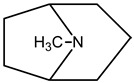
2.	Pyrrolizidine alkaloids	Asteraceae Fabaceae	N-oxides Senecionine	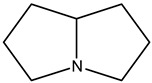
3.	Piperidine alkaloids		Piperidine nucleus Monocycle compounds C5N nucleus	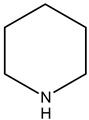
4.	Quinolines alkaloid	Cinchona plant	Quinolone-nucleus Cinchonine, Cinchonidine, Quinine Quinidine	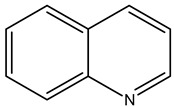
5.	Isoquinoline alkaloids	Higher plants	Morphine Codeine Salsoline, Mimosamycin Reticuline Dauricine Imbricatine	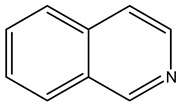
6.	Indole alkaloids		Tryptophan Aplysinopsin Tryptamine Gramine Rutaecarpine Canthin-6-One Ergotamine	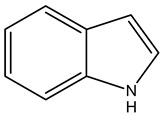
7.	Steroidal alkaloids	Liliaceae Apocynaceae Solanaceae Buxaceae	Veratramine Cyclopamine Solanidine Tomatidine Solasodine	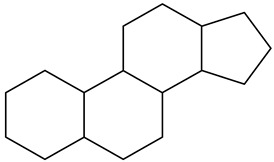
8.	Imidazole alkaloid	Rutaceae	Imidazole ring pharmaceutical potential Pilocarpine	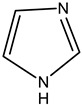
9.	Purine alkaloids	Sapindaceae	Caffeine Theophylline Theobromine	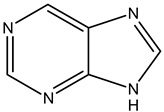
10.	Pyrrolidine alkaloids	Fabaceae Boraginaceae Eupatorium Asteraceae	Pyrrolidine (C_4_N skeleton) Hygrine Ficine Brevicolline	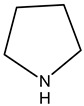

**Table 3 cimb-47-00030-t003:** Potential pharmacological benefits of Glycosides.

Compound	Source	Potential	Reference
Paradoxoside A	*Vitellaria paradoxa*	Anticancer Potential	[14,15,16]
Butyroside	*Vitellaria paradoxa*		
Tieghemelin	*Vitellaria paradoxa*		
papyriferoside A	*Betula papyrifera* *Betula papyrifera*		
5-O-β-D-apiofuranosyl-(1→2)-β-D-glucopyranosyl-1, 7-bis-(4-hydroxyphenyl)-heptan-3-one	*Betula papyrifera*		
Platyphyl Loside	*Betula papyrifera*		
Solamargine	*Solanum incanum*		
Antiaroside J	*Antiaris toxicaria*	Toxic potential against human NIHH460 lung cancer cells	[13,17,18]
Antiaroside N	*Antiaris toxicaria*		
Antiaroside O	*Antiaris toxicaria*		
Antiaroside P	*Antiaris toxicaria*		
Antiaroside	*Antiaris toxicaria*		
Deglucocheirotoxol	*Antiaris toxicaria*		
Convallatoxol	*Antiaris toxicaria*		
Desglucocheirotoxin	*Antiaris toxicaria*		
Strophalloside	*Antiaris toxicaria*		
Convallatoxin	*Antiaris toxicaria*		
Toxicarioside B	*Antiaris toxicaria*		
Antialloside	*Antiaris toxicaria*		
Antiarin	*Antiaris toxicaria*		
Antiaroside Q	*Antiaris toxicaria*		

**Table 4 cimb-47-00030-t004:** Major phytochemicals and breast cancer.

Phytochemicals	Cancer Target	Therapeutic Effect	Reference
*Phenolics*, *terpenoids*, and *alkaloids*	MCF-7 and MDA-MB-231	Antitumoral activity	[36]
*Butea monosperma extracts*	MCF-7 breast cancer cell	Apoptosis in MCF-7 breast cancer cells Reducing mitochondrial membrane potential Inhibition of MCF-7 cells by arresting cell cycle	[37]
*Bulbine frutescens* *phytochemicals*	T47D cells of breast cancer	Increase ROS production Inhibit Notch signaling pathway	[38]
*Polyphenols*, *Quercetin*, *curcumin*, and *resveratrol*	Cancer Stem Cells (CSCs)	Overcome drug resistance	[30]
*β-terpineol*, *1*,*5-cyclooctadiene*, *3-(methyl-2)propenyl*, and *cyclohexene*	MDA-MB-231	Induced apoptosis Cytotoxic and antitumoral activity	[39]
*Quercetin*	MCF-7 breast cancer cell	Inhibition of cell cycle progression	[40]
*Polyphenols*	Breast Cancer Cells (MDA-MB-231)	Phosphatidylinositol 3-kinase (PI3K)/AKT content was decreased NFκB activation in MDA-MB-231 breast cancer cells	[41]
*Flavonols*, *flavan-3-ols* and *anthocyanidins*	Phytoestrogens analysis	Reduction in breast cancer risk	[42]
*Glycitein*	Phytoestrogens analysis	Reduction in breast cancer risk	[43]

**Table 5 cimb-47-00030-t005:** Major compounds with potent role in breast cancer.

Compound	Target	Structure	Mechanism	Reference
Genistein	Breast cancer	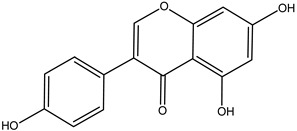	Apoptosis Bax/Bcl-2 G2/M phase arrest	[79,80,81,82]
Quercetin		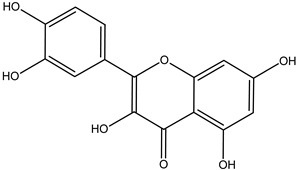	Apoptosis Activation of caspase-3 Activation of caspase-9	[83,84]
Parthenolide		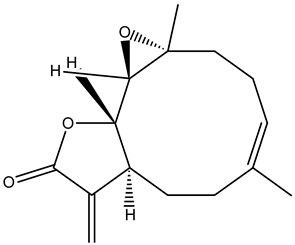	Cell-cycle arrest	[85]
Xanthatin		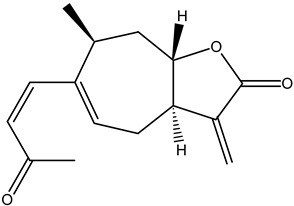	N-acetyl-L-cysteine (NAC)-sensitive mechanism Human breast cancer MDA-MB-231 cells	[86,87,88]
Rosmanol		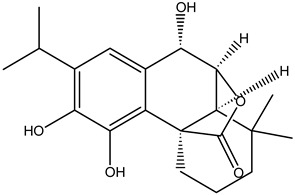	Regulating PI3K/AKT signaling pathways Regulating STAT3/JAK2 signaling pathways	[89]

**Table 6 cimb-47-00030-t006:** Major compounds with concentration.

Compound/Extract/Plant Extract	Method	Target	Concentration (IC_50_)	References
*A. muricata Methanol extract*	Extract	MDA-MB-231 and MCF7 cell lines.	17 and 23 µg/ml	[91]
*A. muricata water extract*	Extract	MDA-MB-231 and MCF7 cell lines.	19 and 24.5 µg/ml	[91]
*ASE*	Extract	MCF-7 cells	1400 µg/ml	[91]
*Carotenoids*	Fractionation	MDA-MB-231	4.25 µg/ml	[92]
*Squalene*	Fractionation	MDA-MB-231	16.8 µg/ml	[92]
*Fraction of S. crispus*	Fractionation	MCF-7 cells	100 µg/ml	[93]
*Bryonia dioica*	Extract	MCF-7	9.81 mg/mL	[20]
*Baeckea frutescens*	Plant	MCF-7 MDA-MB-231, MCF10A	53 μg/mL	[20]
*Bulbine frutescens*	Plant	MDA-MB-231, T47D	4.8–28.4 μg/mL	[20]
*Cimicifuga dahurica*	Plant	MCF-7	30 μM	[20]
*Fagonia indica*	Plant	MCF-7, MDA-MB-468	50–100 μM	[20]
*Glycyrrhiza glabra*	Plant	Multiple cell line	0 or 20 mg/kg	[20]
*Lawsonia nermis*	Plant	MCF-7	1.5 μM	[20]
*Morus alba*	Plant	MCF-7	350 μg/mL	[20]
*Premna odorata*	Plant	MCF-7, BT-474	13.3 μM	[20]
*Salvia species*	Plant	T47D, ZR-75-1, BT 474	30 μg/mL	[20]
*Senecio graveolens*	Plant	ZR-75-1, MDA-MB-231	200 μg/mL	[20]
EVO	Isolation	MDA-MB-231	17.48 μg/mL	[94]
ENPs	Isolation	MCF-7	7.86 μg/mL	[94]
oleanolic acid	Compound	DA-MB-231 cells	28.02 µg/mL	[95]
β-sitosterol	Compound	MCF-7	15.42 µg/mL	[95]
β-amyrin	Compound	MCF-7	10.08 µg/mL	[95]
β-sitosterol-glucoside	Compound	MCF-7	11.34 µg/mL	[95]
(22E, 24S)-Ergosta-4,6,8(14), 22-tetraen-3-one	Compound	MCF-7	24.2 µM	[96]
β-sitosterol s	Compound	MCF-7	24.6 µM	[96]
Walsucochinone C	Compound	MCF-7	16.4 µM	[96]
Nimonol	Compound	MCF-7	22.03 µM	[96]
Limonoid Kihadanin B	Compound	MDA-MB-231	7.79 µM	[97]
Chisopaten (A-D)	Compound	MCF-7	4.01 µM	[98]
Chisopaten (A-D)	Compound	MCF-7	4.33 µM	[98]
melianodiol	Compound	MCF-7	16.84 µM	[99]
cycloschimperols B	Compound	MCF-7	2.10 µM	[99]
neriifolins A	Compound	MCF-7	13.14 µM	[100]
neriifolins B	Compound	MCF-7	7.12 µM	[100]
neriifolins C	Compound	MCF-7	9.50 µM	[100]
Betulinic acid (BA)	Compound	MCF-7	19.06 µM	[101]
Cucurbitacin B (CuB)	Compound	MDA-MB-231	15.89 µM	[102]
Cucurbitacin B (CuB)	Compound	SKBR-3 cells	6.177 µM	[102]
3-O-(E)-p-coumaroylbetulinic acid (CB)	Compound	MDA-MB-231	5.884 µM	[103]
3-O-(E)-p-coumaroylbetulinic acid (CB)	Compound	T-47D cells	2.708 µM	[103]

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
