# Peer review of "Phytochemicals in Breast Cancer Prevention and Treatment: A Comprehensive Review"

_cimb, 2025, doi:10.3390/cimb47010030_

Round 1
Reviewer 1 Report
Comments and Suggestions for Authors
The manuscript presents a comprehensive review of the role of natural products, specifically phytochemicals, in breast cancer prevention and treatment. The authors discuss various classes of phytochemicals, including flavonoids, terpenoids, polyphenols, and glycosides, and their impact on cancer pathways such as JAK/STAT3, HER2-integrin, and MAPK. Additionally, the manuscript evaluates advanced delivery methods, including liposomes and nanoemulsions, as potential enhancements for phytochemical efficacy. The paper provides valuable insight into the complementary role of phytochemicals in addressing the challenges of current cancer treatments, such as toxicity and resistance, but requires more depth and clarity in certain areas to ensure scientific rigor.
Major Comments:
1.Many of the claims regarding the efficacy of phytochemicals, particularly their effects on cancer pathways like MAPK and JAK/STAT3, rely heavily on secondary literature. To strengthen the manuscript, the authors should include more primary experimental evidence, such as in vitro or in vivo studies, to validate these claims. Providing specific examples with quantitative data or experimental results would help support these assertions and improve the credibility of the manuscript.
2. The discussion of molecular pathways, such as HER2-integrin and MAPK, would benefit from more detailed explanations regarding the biochemical interactions of the phytochemicals with these pathways. The current figures illustrating these pathways are insufficiently labeled and lack clarity. Adding more comprehensive diagrams or flowcharts with detailed annotations would greatly enhance reader comprehension.
3. Although the manuscript addresses the synergistic potential of phytochemicals with conventional therapies, it lacks sufficient discussion on the challenges of translating these findings into clinical practice. Issues like low bioavailability, variability in plant extract composition, and the risk of adverse effects require further attention. Including references to ongoing or completed clinical trials, as well as strategies to address these challenges, would provide a more balanced and realistic perspective.
4. While the concept of personalized phytochemical therapies is briefly introduced, the authors should explore how such treatments could be developed using genomic profiling or biomarkers. A deeper discussion on personalized treatment strategies and their future applications could add significant value to the manuscript.
Minor Comments:
1. The quality of figures should be enhanced for better resolution, and all abbreviations should be clearly defined in the figure legends. Additionally, color schemes should be adjusted for better visibility.
2. Terminology throughout the manuscript is inconsistent, such as alternating between "phytochemicals" and "bioactive compounds." Standardizing terms would improve readability and prevent confusion.
3. Certain sentences would benefit from grammatical refinement for better readability. For instance, rephrasing "poisonous potentials induced" to "toxicity induced" would make the statement clearer.
4. The reference list should be reviewed to ensure consistency with the journal's formatting guidelines. In-text citations should also be standardized.
5. Expanding the list of keywords to include terms such as "nanotechnology," "targeted drug delivery," and "synergistic effects" could improve the manuscript's discoverability in relevant searches.
Author Response
Comments and Suggestions for Authors
The manuscript presents a comprehensive review of the role of natural products, specifically phytochemicals, in breast cancer prevention and treatment. The authors discuss various classes of phytochemicals, including flavonoids, terpenoids, polyphenols, and glycosides, and their impact on cancer pathways such as JAK/STAT3, HER2-integrin, and MAPK. Additionally, the manuscript evaluates advanced delivery methods, including liposomes and nanoemulsions, as potential enhancements for phytochemical efficacy. The paper provides valuable insight into the complementary role of phytochemicals in addressing the challenges of current cancer treatments, such as toxicity and resistance, but requires more depth and clarity in certain areas to ensure scientific rigor.
Major Comments:
1.Many of the claims regarding the efficacy of phytochemicals, particularly their effects on cancer pathways like MAPK and JAK/STAT3, rely heavily on secondary literature. To strengthen the manuscript, the authors should include more primary experimental evidence, such as in vitro or in vivo studies, to validate these claims. Providing specific examples with quantitative data or experimental results would help support these assertions and improve the credibility of the manuscript.
Response: As suggested by the learned reviewer, we have in cooperated more example of experimental evidence to validate the clam.
- The discussion of molecular pathways, such as HER2-integrin and MAPK, would benefit from more detailed explanations regarding the biochemical interactions of the phytochemicals with these pathways. The current figures illustrating these pathways are insufficiently labeled and lack clarity. Adding more comprehensive diagrams or flowcharts with detailed annotations would greatly enhance reader comprehension.
Response: As suggested by the learned reviewer, we have included labeled figures that depict these pathways.
- Although the manuscript addresses the synergistic potential of phytochemicals with conventional therapies, it lacks sufficient discussion on the challenges of translating these findings into clinical practice. Issues like low bioavailability, variability in plant extract composition, and the risk of adverse effects require further attention. Including references to ongoing or completed clinical trials, as well as strategies to address these challenges, would provide a more balanced and realistic perspective.
Response: We appreciate the reviewer’s insightful comments regarding the challenges of translating phytochemical research into clinical practice. To address this, we have expanded the discussion in the manuscript
- While the concept of personalized phytochemical therapies is briefly introduced, the authors should explore how such treatments could be developed using genomic profiling or biomarkers. A deeper discussion on personalized treatment strategies and their future applications could add significant value to the manuscript.
Response: Thank you for this valuable suggestion. In response, we have expanded the discussion on personalized phytochemical therapies by integrating insights on the role of genomic profiling and biomarkers in their development. Specifically, we have addressed how genomic data can be utilized to identify individual susceptibilities and optimize phytochemical interventions tailored to specific genetic and phenotypic profiles.
Additionally, we have included an exploration of emerging technologies, such as metabolomics and proteomics, that complement genomic profiling in advancing personalized treatments. We also discussed potential future applications, including the integration of bioinformatics and AI tools in the design of phytochemical therapies that align with individual health needs. These additions can now be found in the revised manuscript using track change. We trust that this expanded discussion enhances the value of the manuscript and addresses the reviewer’s suggestion.
Minor Comments:
1. The quality of figures should be enhanced for better resolution, and all abbreviations should be clearly defined in the figure legends. Additionally, color schemes should be adjusted for better visibility.
Response: As suggested by the learned reviewer, we have enhanced the resolution of all figures to ensure better clarity and visual quality. All abbreviations have been clearly defined in the respective figure legends to ensure they are understandable to the readers. The color schemes of the figures have been adjusted for improved visibility and contrast, making them more reader-friendly.
- Terminology throughout the manuscript is inconsistent, such as alternating between "phytochemicals" and "bioactive compounds." Standardizing terms would improve readability and prevent confusion.
Response: We sincerely appreciate the reviewer’s observation regarding the inconsistent use of terminology throughout the manuscript. To address this, we have carefully reviewed the manuscript and standardized the terminology. We have chosen to use the term “bioactive compounds” consistently throughout the text, as it encompasses both phytochemicals and other relevant compounds discussed. This revision has been applied to all sections of the manuscript to ensure clarity and improve readability.
- Certain sentences would benefit from grammatical refinement for better readability. For instance, rephrasing "poisonous potentials induced" to "toxicity induced" would make the statement clearer.
Response: Thank you for your valuable feedback. We appreciate your suggestion to refine certain sentences for better readability. We have revised the phrase "poisonous potentials induced" to "toxicity induced" as recommended, to enhance clarity and improve the flow of the text. We believe this adjustment strengthens the manuscript and improves its overall readability.
- The reference list should be reviewed to ensure consistency with the journal's formatting guidelines. In-text citations should also be standardized.
Response: Thank you for highlighting this issue. We have thoroughly reviewed the reference list and formatted it according to the journal's guidelines. All in-text citations have also been standardized to ensure consistency with the prescribed formatting requirements. We appreciate your attention to detail and believe these revisions have enhanced the overall presentation of the manuscript.
- Expanding the list of keywords to include terms such as "nanotechnology," "targeted drug delivery," and "synergistic effects" could improve the manuscript's discoverability in relevant searches.
Response: Thank you for your valuable suggestion to expand the list of keywords. We have incorporated the terms "nanotechnology," "targeted drug delivery," and "synergistic effects" into the manuscript's keywords. These additions will indeed enhance the discoverability of our work in relevant searches and align well with the scope of the study
Reviewer 2 Report
Comments and Suggestions for Authors
Dear authors,
I have now read your submission with ID cimb-3356443. While the content is thorough and informative, it does have issues with redundancy, clarity, and depth of analysis which limit its overall impact. My comments are as follows:
1. The current title reads so redundant. It could be rephrased to: Phytochemicals in Breast Cancer Prevention and Treatment: A Comprehensive Review.
2. Abstract: This should be revised to clearly indicate a brief background, aim of the review and the key findings. Any directions for future research should be indicated to help guide the field further.
3. Please don’t repeat words that are in the title of the manuscript as keywords.
4. Please streamline sections that discuss overlapping concepts (for example, signaling pathways and synergistic effects) to improve conciseness and focus. I also do not see the relevance of section 1.2 (L97-113); maybe it could be merged entirely with the INTRODUCTION.
5. L115-162: This should be under INTRODUCTION. Please try to summarize it as much as possible. There is no need for the subsections that could mislead the readers. For instance, section 2.1 is on definition but it also provides prevalence of breast cancer.
6. The introduction needs to start from a broader definition of what cancer is, its prevalence and the treatment challenges before highlighting the prevalence of breast cancer. The text should then conduct the reader into why phytochemicals are being sought for cancer treatment.
7. In Table 1, revise: Ttropane (C4N skeleton) >> Tropane (C4N skeleton).
8. Please improve the integration of figures (especially the signaling pathways) with the main text to ensure they complement the discussion effectively. Consider adding explanatory legends for clarity. Some words in figure 1 are not readable.
9. In the main text, please critically discuss limitations of using phytochemicals, including gaps in clinical trials, issues with bioavailability, and potential adverse effects (toxicity) of phytochemicals.
10. The conclusion could be strengthened by summarizing the most critical findings and proposing actionable recommendations for researchers and clinicians.
Author Response
- The current title reads so redundant. It could be rephrased to: Phytochemicals in Breast Cancer Prevention and Treatment: A Comprehensive Review.
Response: As suggested by the learned reviewer, the title is changed to “Phytochemicals as Key Agents in Breast Cancer Prevention and Treatment: A Comprehensive Review”
- Abstract: This should be revised to clearly indicate a brief background, aim of the review and the key findings. Any directions for future research should be indicated to help guide the field further.
Response: Extensive investigation has been conducted on plant-based resources for their pharmacological usefulness, including for various cancer types. The scope of this review is wider than several studies with a particular focus on breast cancer, which is an international health concern while studying sources of flavonoids, carotenoids, polyphenols, saponins, phenolic compounds, terpenoids, and glycosides apart from focusing on nursing. Important findings from prior studies are synthesized to explore these compounds' sources, mechanisms of action, complementary and synergistic effects, and associated side effects. It was reviewed that the exposure to certain doses of catechins, piperlongumine, lycopene, isoflavones, cucurbitacin, and BPEITC for a sufficient period can provide profound anticancer benefits through biological events such as cell cycle arrest, cells undergoing apoptosis and disruption of signaling pathways including, but not limited to JAK-STAT3, HER2-integrin and MAPK. Besides, the study also covers potential adverse effects of these phyto including the dangers they pose. Regarding mechanisms widest attention is paid to Complementary and synergistic strategies are discussed which indicate that it would be realistic to alter the dosage and delivery systems of liposomes, nanoparticles, nanoemulsions, and films to enhance efficacy. Future research directions include refining these delivery approaches, further elucidating molecular mechanisms, and conducting clinical trials to validate findings. These efforts could significantly advance the role of phytocompounds in breast cancer management.
- Please don’t repeat words that are in the title of the manuscript as keywords.
Response: Anticancer activity; Flavonoids; Apoptosis; Signaling Pathways Inhibition; Synergistic Effects; Delivery Strategies
- Please streamline sections that discuss overlapping concepts (for example, signaling pathways and synergistic effects) to improve conciseness and focus. I also do not see the relevance of section 1.2 (L97-113); maybe it could be merged entirely with the INTRODUCTION.
Response: The anti-cancer activity of numerous medicinal plants is attributable to phytochemicals. All phytochemicals derived from plants, including the Dimocarpus longan, Piper longum, Withania somnifera, Nigella sativa, Curcuma longa, Murraya koenigii, as well as Amora rohituka are of significant importance in drug development. Recently, the focus of research has shifted toward formulating targeted phytochemicals which could be used to relieve the toxic effects of cancer treatment. Across the globe, action is being taken to implement these measures, and this call for a detailed evaluation of such action. This review evaluates the evidence regarding the use of plant chemicals in the prevention and treatment of breast cancers, including working mechanisms of plant chemicals, possible protective effects, dietary sources, in vitro, animal, and clinical trial studies.
- L115-162: This should be under INTRODUCTION. Please try to summarize it as much as possible. There is no need for the subsections that could mislead the readers. For instance, section 2.1 is on definition but it also provides prevalence of breast cancer.
Response: According to the Centers for Disease Control and Prevention (CDC), breast cancer is defined as a disease in which cells grow out of control within the breast. The classification of breast cancer primarily relied on the type of cells which is turned into cancer. As per the recent statistics in 2020 [6], it was inferred that about 2 million new cases were reported in 2020 and the incidence rate has also increased over the past few years.
Breast cancer can be divided into two main categories: non-invasive and invasive. Non-invasive breast cancer, such as Ductal Carcinoma in Situ (DCIS), is confined to ducts and does not spread to adjacent tissues. In contrast, invasive breast cancer spreads to surrounding connective and fatty tissues. Furthermore, breast cancer is a heterogeneous disease, and its treatment depends on the expression of surface markers like hormone receptors (HR) and HER2. The primary subtypes of breast cancer include Hormone Receptor-Positive (HR+) / HER2-Negative, which accounts for about 70% of cases and is treated with hormone therapies like tamoxifen; HER2-Positive Breast Cancer, which is marked by the overexpression of HER2 protein and accounts for 15-20% of cases, and is treated with HER2-targeted therapies such as trastuzumab; and Triple-Negative Breast Cancer (TNBC), which lacks ER, PR, and HER2 expression and is often more aggressive, requiring chemotherapy as the main treatment, with ongoing research into alternative therapies [7-9]. The identification of risk factors for breast cancer is essential for effective screening and prevention. Seven key risk factors are associated with an increased likelihood of developing breast cancer: age, gender, personal and family history of breast cancer, histologic risk factors, reproductive factors, exogenous hormone use, and genetic predisposition. Notably, the risk of breast cancer rises with age, and individuals with first-degree relatives who have the disease face a 2-3 times higher risk of developing it [10].
- The introduction needs to start from a broader definition of what cancer is, its prevalence and the treatment challenges before highlighting the prevalence of breast cancer. The text should then conduct the reader into why phytochemicals are being sought for cancer treatment.
- In Table 1, revise: Ttropane (C4N skeleton) >> Tropane (C4N skeleton).
Response: Tropane
- Please improve the integration of figures (especially the signaling pathways) with the main text to ensure they complement the discussion effectively. Consider adding explanatory legends for clarity. Some words in figure 1 are not readable.
Response: Figure 1: The diagram explicates the molecular mechanisms of phytochemicals in cancer treatment. It demonstrates their roles on cellular proliferation, apoptosis, and cell cycle. Phytochemicals prevent angiogenesis, metastasis, oxidative stress, and inflammation, and regulate redox signaling. They inhibit enzymes, modulate the mammosphere formation, and enhance immune system activity. The anti-apoptotic modulation and targeting of breast cancer cells and preventing the rate of perfusion looks positive due to the antioxidative nature of the flavonoids.
Figure 2: The figure illustrates key signaling pathways involved in cancer progression and drug resistance:
(A) Akt/PI3K/mTOR Pathway: Growth factors activate receptor tyrosine kinases (RTKs), stimulating PI3K and Akt, leading to mTORC1 activation. This regulates cell growth, proliferation, apoptosis, metastasis, and drug resistance by modulating PTEN expression, miR-21 levels, 4E-BP1, and S6k1.
(B) MAPK Pathway: RTK activation triggers RAS, leading to RAF and MEK activation. Downstream effectors (JNK, p38, ERK 1/2) influence similar cellular processes.
(C) JAK/STAT Pathway: Cytokines stimulate JAK, phosphorylating STATs, which translocate to the nucleus, driving genes involved in growth and drug resistance.
- In the main text, please critically discuss limitations of using phytochemicals, including gaps in clinical trials, issues with bioavailability, and potential adverse effects (toxicity) of phytochemicals.
Response: The discussion on the limitations of phytochemicals, including the gaps in clinical trials, challenges with bioavailability, and potential adverse effects (such as toxicity), has been in cooperated into the revised manuscript with tracked changes.
- The conclusion could be strengthened by summarizing the most critical findings and proposing actionable recommendations for researchers and clinicians.
Response: We have included a summary of the key findings and offered practical recommendations for researchers and clinicians in the conclusion section.
Reviewer 3 Report
Comments and Suggestions for Authors
The manuscript aims to comprehensively review phytochemicals and their potential applications in breast cancer prevention and treatment. It covers a broad range of topics, from the mechanisms of action of phytochemicals to their integration with conventional therapies and delivery systems. While the work is ambitious and holds promise, significant revisions are required to meet publication standards.
Minor:
1. Improve the alignment of tables and ensure clear legends or explanatory notes accompany them. Their current state makes it challenging to interpret the data.
2. Replace low-quality figures with high-resolution images and ensure all figures are appropriately labelled and referenced within the text.
3. Ensure that section headings and subheadings follow a uniform structure
4. Eliminate repetitive discussions, especially regarding phytochemical classifications and mechanisms of action, to streamline the narrative.
5. Expand the conclusion section to succinctly capture the findings and highlight the gaps that future research could address
Major:
1. The manuscript attempts to cover an extensive range of topics, compromising the depth of analysis. Splitting the work into two manuscripts—one focusing on phytochemical mechanisms and their biological effects and the other on delivery strategies and clinical implications—would provide more clarity and impact.
2. While the manuscript includes detailed reviews of phytochemical mechanisms, it lacks a thorough discussion of their clinical translation. Address the challenges and opportunities associated with using these compounds in real-world settings.
3. Highlight specific examples of clinical trials, successful applications, or case studies where phytochemicals have demonstrated efficacy in breast cancer treatment.
4. Add a dedicated section that outlines clear, actionable steps for future research, such as developing standardized dosages, improved delivery systems, and personalized medicine approaches.
5. Discuss potential synergies with conventional therapies in greater depth and explore how these could mitigate the limitations of current cancer treatments.
The manuscript shows ambition and covers a vital area of research, but its presentation and focus need improvement before publication. The effort to synthesize diverse aspects of breast cancer treatment with phytochemicals is commendable, but more transparent structure and refinement are essential.
Comments on the Quality of English Language
Minor stylistic and linguistic corrections are required
Author Response
The manuscript aims to comprehensively review phytochemicals and their potential applications in breast cancer prevention and treatment. It covers a broad range of topics, from the mechanisms of action of phytochemicals to their integration with conventional therapies and delivery systems. While the work is ambitious and holds promise, significant revisions are required to meet publication standards.
Minor:
- Improve the alignment of tables and ensure clear legends or explanatory notes accompany them. Their current state makes it challenging to interpret the data.
Response: We sincerely appreciate your constructive feedback regarding the alignment of tables and the need for clear legends or explanatory notes. We have thoroughly revised the tables to improve their alignment for better visual presentation and readability. Additionally, we have included detailed legends and explanatory notes for each table to ensure the data is easy to interpret and provides sufficient context.
- Replace low-quality figures with high-resolution images and ensure all figures are appropriately labelled and referenced within the text.
Response: Thank you for your valuable feedback. We have addressed this comment by replacing all low-quality figures with high-resolution images to enhance the clarity and visual impact. Additionally, we have ensured that all figures are appropriately labelled and referenced within the text. Please find the revised figures incorporated in the manuscript, and the updated labels and references are highlighted in the revised version for your review.
- Ensure that section headings and subheadings follow a uniform structure
Response: Thank you for pointing this out. We have carefully reviewed the manuscript and ensured that all section headings and subheadings are consistent in style and format throughout the document.
- Eliminate repetitive discussions, especially regarding phytochemical classifications and mechanisms of action, to streamline the narrative.
Response: We appreciate the valuable feedback provided regarding the elimination of repetitive discussions, particularly those concerning phytochemical classifications and mechanisms of action. We have carefully reviewed the manuscript and identified sections where redundancy occurred. These sections have now been streamlined to ensure a more concise narrative.
- Expand the conclusion section to succinctly capture the findings and highlight the gaps that future research could address
Response: Thank you for your valuable feedback and for highlighting the need to expand the conclusion section. We have revised the conclusion to succinctly summarize the key findings of our review and explicitly outline the research gaps that warrant further exploration.
Major:
- The manuscript attempts to cover an extensive range of topics, compromising the depth of analysis. Splitting the work into two manuscripts—one focusing on phytochemical mechanisms and their biological effects and the other on delivery strategies and clinical implications—would provide more clarity and impact.
Response: Thank you for your insightful feedback regarding the scope and depth of our manuscript. We appreciate your suggestion to split the work into two manuscripts for greater clarity and impact.
Our intention was to provide a comprehensive overview that connects the phytochemical mechanisms and their biological effects with delivery strategies and clinical implications. We believe this integrative approach highlights the translational potential of these natural compounds, which is a key strength of the manuscript.
However, we understand your concern regarding the breadth of the topics covered. To address this:
- We propose refining certain sections to enhance focus and depth, particularly on the mechanisms and biological effects of the phytochemicals.
- Additionally, we will streamline the discussion on delivery strategies and clinical implications to ensure a cohesive narrative while maintaining the manuscript’s comprehensive scope.
Alternatively, if the editorial team believes splitting the manuscript would better serve the journal’s audience, we are open to revising the work into two distinct submissions as per your recommendation. Please let us know your preference so that we can proceed accordingly. Thank you once again for your valuable suggestions.
- While the manuscript includes detailed reviews of phytochemical mechanisms, it lacks a thorough discussion of their clinical translation. Address the challenges and opportunities associated with using these compounds in real-world settings.
Response: We thank the reviewer for this insightful comment. Recognizing the importance of bridging phytochemical mechanisms with clinical application, we have revised the manuscript to address the challenges and opportunities associated with translating these compounds into real-world clinical settings.
- Highlight specific examples of clinical trials, successful applications, or case studies where phytochemicals have demonstrated efficacy in breast cancer treatment.
Response: We appreciate your insightful comment and have revised the manuscript to include specific examples of clinical trials, successful applications, and case studies demonstrating the efficacy of phytochemicals in breast cancer treatment.
- Add a dedicated section that outlines clear, actionable steps for future research, such as developing standardized dosages, improved delivery systems, and personalized medicine approaches.
Response: Thank you for your valuable feedback. We appreciate your suggestion to add a dedicated section outlining actionable steps for future research. In response to your recommendation, we have included a new section in the manuscript that highlights key areas for future investigation.
- Discuss potential synergies with conventional therapies in greater depth and explore how these could mitigate the limitations of current cancer treatments.
Response: In response to the reviewer's comment on discussing potential synergies with conventional therapies in greater depth, we appreciate the suggestion and have expanded on the topic in the revised manuscript.
Round 2
Reviewer 2 Report
Comments and Suggestions for Authors
Dear authors,
Your review has been refined but I still have some suggestions in the attachment.
Please ensure that your references are aligned with the journal guidelines

Could be improved for a smooth flow
Author Response
Thank you for your valuable feedback and suggestions. We have carefully addressed all the points raised in your review. The necessary revisions have been incorporated into the manuscript, and we have used track changes to highlight these updates for your convenience.
Additionally, we have ensured that all references are now fully aligned with the journal's guidelines.
We appreciate your time and effort in reviewing our manuscript and hope that the revisions meet your expectations. Please do not hesitate to let us know if further adjustments are required.